# Pathogenesis of Keratinocyte Carcinomas and the Therapeutic Potential of Medicinal Plants and Phytochemicals

**DOI:** 10.3390/molecules26071979

**Published:** 2021-04-01

**Authors:** Andrea Jess Josiah, Danielle Twilley, Sreejarani Kesavan Pillai, Suprakas Sinha Ray, Namrita Lall

**Affiliations:** 1DST-CSIR National Centre for Nanostructured Materials, Council for Scientific and Industrial Research, Pretoria 0001, South Africa; andreajess.josiah@gmail.com (A.J.J.); skpillai@csir.co.za (S.K.P.); 2Department of Chemical Sciences, University of Johannesburg, Doornfontein 2028, South Africa; 3Department of Plant and Soil Sciences, Faculty of Natural and Agricultural Sciences, University of Pretoria, Pretoria 0002, South Africa; berrington.danielle@gmail.com (D.T.); namrita.lall@up.ac.za (N.L.); 4School of Natural Resources, University of Missouri, Columbia, MO 65211, USA; 5College of Pharmacy, JSS Academy of Higher Education and Research, Mysuru 570015, India

**Keywords:** keratinocyte carcinoma, medicinal plants, phytochemistry, transdermal drug delivery systems

## Abstract

Keratinocyte carcinoma (KC) is a form of skin cancer that develops in keratinocytes, which are the predominant cells present in the epidermis layer of the skin. Keratinocyte carcinoma comprises two sub-types, namely basal cell carcinoma (BCC) and squamous cell carcinoma (SCC). This review provides a holistic literature assessment of the origin, diagnosis methods, contributing factors, and current topical treatments of KC. Additionally, it explores the increase in KC cases that occurred globally over the past ten years. One of the principal concepts highlighted in this article is the adverse effects linked to conventional treatment methods of KC and how novel treatment strategies that combine phytochemistry and transdermal drug delivery systems offer an alternative approach for treatment. However, more in vitro and in vivo studies are required to fully assess the efficacy, mechanism of action, and safety profile of these phytochemical based transdermal chemotherapeutics.

## 1. Introduction

Cutaneous carcinoma, or skin cancer, remains one of the highest occurring cancer types, with the number of incidences increasing globally. Although the number of cases differs significantly depending on the geographical region, it remains a major health care problem across the world, as it affects both men and women of every ethnicity/race [1].

There are two criteria that classify skin cancer; the cell from which they originate and their clinical behavior. Skin cancer is categorized into two major groups: non-melanoma skin cancer (NMSC) and malignant melanoma. The cases of non-melanoma skin cancer are generally substantially higher than melanoma cases and are often much less complex to treat compared to melanoma, as it has a lower metastatic potential [2]. Non-melanoma skin cancer, consisting of basal cell carcinoma (BCC) and squamous cell carcinoma (SCC), is often referred to as keratinocyte carcinoma, as these two subtypes are derived from the epidermal keratinocytes [3].

## 2. Methodology

A scientific literature search was performed during 2018–2020, using several databases (Science Direct, Google Scholar, Scopus, and Pubmed). The literature search focused on the following topics; KC, BCC, and SCC. Review articles linked to these topics contributed extensively to the structure of the current review. To understand the severity of KC and the increase in cases and mortality, various research papers, and cancer registries were analyzed. Previous research papers that reported on in vitro and in vivo studies regarding the use of medicinal plants, phytochemicals, and transdermal drug carriers provided insights on novel treatments for KC. Literature that was reviewed ranged from the year 1987–2020, which included research articles, patents, book chapters, and review articles.

## 3. The Origin and History of Keratinocyte Carcinoma

Basal cell carcinoma and SCC originate within the epidermis layer of the skin [4]. The epidermis layer undergoes a process known as homeostasis, where new cells replace old or damaged cells. Homeostasis and the regeneration of skin cells are maintained by stem cells located within the epithelial tissue, which have the ability to differentiate into different cell lineages and are able to self-renew. The epidermis layer is comprised of three main compartments: the interfollicular epidermis, the hair follicle, and the sebaceous and sweat glands [5]. There have been numerous conflicting results on the epidermal lineage of BCC and SCC cells, which are discussed below.

### 3.1. Basal Cell Carcinoma

The origin of BCC has been debated since the early 1900s. Krompecher defined the origin of BCC as a tumor that arose from the basal cell layer of the interfollicular epidermis [6]. Several researchers over the years have endorsed Krompechers’ definition. However, Lever [7] had a contrary view, which explained that BCC is of follicular origin, derived from an epithelial hair germ. Immunohistochemical studies have been conducted that substantiated Levers’ view [8]. A study conducted by Van Scott and Reinertson in the year 1961, discovered that the growth of tumorous epithelial cells is dependent on their stroma. A later study validated the findings by Van Scott and Reinerston and further reported on the stromal properties, which are fundamental requirements for tumor growth, which include platelet-derived growth factor (PDGF) A and B, and their corresponding receptors α and β. Another study that focused on gene expression profiling of BCC stromal tissue demonstrated that PDGF receptor-like protein displayed an upregulation in BCC stroma. Extensive studies are necessary to accurately assess the effect of the stroma in relation to BCC tumor growth [9,10].

In the year 1824, Aurther Jacob, a member of the royal college of surgeons in Ireland, first described what we now term “basal cell carcinoma”. He conducted immunohistochemical studies that led him to define BCC as a mass of cancerous cells, which were germinative keratinocytes that emerged from hair follicles [11]. Further research conducted by a German pathologist, Krompecher Odon, in 1900 validated the discovery formed by Jacob. In the year 1903, Krompecher published a book called Der Basalzellenkrebs (The Basal Cell Cancer), which explained that these specific tumors arise from the lowest layer of cells present in the skin [6]. A recent study performed by Tan et al. confirmed that BCC originates within the basal layer of the epidermis and arises from the interfollicular epidermis [12].

### 3.2. Squamous Cell Carcinoma

A review by Kipling et al. provided a fundamental timeline that established the discovery of SCC. Heinrich Bass (Bassius) first described scrotum carcinoma in 1731, which was published in an article entitled ‘*Scrotum sphacelo consumptum et fenatum*’. In the year 1740, Treyling also confirmed the description of scrotum carcinoma in an article titled ‘*Scrotum immaniter auctum scirrhoso scrophulosm*’ [13]. However, Percivall Pott, an English surgeon and scientist, was the first to assign occupational cause to the disease. In the year 1775, Pott established that high incidence rates of scrotal cancer were related to chimney soot exposure. Subsequent studies termed this cancer chimney sweeps carcinoma, which can be described as an extremely rare sub-division of SCC. Potts’ discovery was the inception of a plethora of research articles focused on SCC [14,15]. Squamous cell carcinoma arises from the squamous cell layer, which is located in the epidermal layer of the skin [16]. Therefore, SCC can be described as an epithelial malignancy that occurs in organ covered squamous epithelium [17].

## 4. Incidence and Demographics of Keratinocyte Carcinoma

A report by the World Health Organization (WHO) reveals there is are estimated 2–3 million cases of KC cancer that occur globally each year [18]. Due to the incidence of skin cancer not always being reported, the estimated number of cases that occur globally each year are severely underestimated. It is also evident that there is a large variation in the number of reported cases that occur; therefore, the worldwide burden of KC remains unclear. The occurrence rate of KC in the United States (US) documented by the American Cancer Society revealed that an estimated 3.3 million people were diagnosed with KC; however, the number of cases is severely underestimated, as KC cases are not required to be reported to cancer registries [19]. A report by Rogers et al. substantiated the increase in the number of cases, where 5,434,193 million KC cases were diagnosed, and 3,315,554 individuals were treated in the US in 2015 [20]. According to GLOBOCAN, the estimated number of new KC cases in 2018, excluding basal cell carcinoma cases, was 1,042,056 with 65,155 deaths [21]. Estimates for the number of non-melanoma skin cancer incidence worldwide for 2020 were updated by Globocan (Figure 1) [22].

Basal cell carcinoma is the world’s most prevalent human cancer and is accountable for approximately 70–80% of all skin cancers [23,24]. A report by Dessinioti et al. revealed that Australia had the highest incidence of BCC (~1–2%) annually, with approximately 2448 of a 100,000 population diagnosed in 2011, followed by the US (450 per 100,000 in 2010) and Europe (220.1 per 100,000 yearly average) [25,26]. The South African National Cancer Registry reported approximately 14,414 BCC and 6950 SCC cases in 2016 [27]. Another study found that the lowest incidence rates of BCC were in Finland, in comparison to other European countries [28]. This trend is followed closely by Italy, with 88 out of 100,000 people diagnosed with BCC [29]. Squamous cell carcinoma has been reported to contribute approximately 20% of skin cancer cases [30]. Numerous reports indicate a significant incline in SCC incidence rates over the past three decades, with an approximately yearly increase of 3–10% [26]. Currently, SCC incidence rate is estimated to be between 15–35/100,000 people per year and is set to increase between 2–4% annually, due to chronic ultraviolet B (UVB) exposure and an aging population [31].

## 5. Diagnosis of Keratinocyte Carcinoma

The early diagnosis of KC, such as SCC, is crucial in controlling the risk of cancer becoming invasive and assists in minimizing the complexity of treatment plans. Approximately 70% of all BCCs are found on the head and neck, which can have detrimental effects such as facial deformities and the collapse of certain vital structures [32].

Dermatoscopy, an in vivo imaging diagnostic technique, is used to examine pigmented skin lesions under high magnification in order to distinguish between pigmented BCC and melanoma. Computer tomography or magnetic resonance imaging are used to determine the extent of abnormal cell tissue growth in cartilage, bone, large nerve, eyeball, or parotid gland, whereas skin biopsy aids in identifying KC subtypes and subsequent treatment plans [33]. Lesions that appear clinically abnormal can be tested via biopsy or excision. Abnormal lesions can be examined using incisional biopsy (incision, punch, or shave biopsy), where a section of tissue is removed, or excisional biopsy, where the entire lesion or tissue is removed. Appropriate treatment and prognosis is therefore based on tumor differentiation grade, histologic subtype, degree of dermal invasion, tumor depth, presence or absence of perineural, lymphatic, or vascular invasion, and extension of tumor cells to margins [34,35]. A tumor staging system, such as the Cancer Staging Manual, published by the American Joint Committee on Cancer (AJCC), is used to determine the prognosis and treatment plant based on the location and size of the tumor, whether it has spread to the lymph nodes, and the extent to which it has spread to other parts of the body [34]. There are two types of staging systems, the number staging system (Table 1) or the TNM (tumor, node, and metastasis) staging system. A full description of the TNM staging system has been described by Fahradyan et al. [35].

## 6. Clinical Variants of BCC

Basal cell carcinoma accounts for approximately 70% of KC cases and most commonly develops on sun-exposed skin, predominantly on the head, but can also develop on the neck, trunk, and lower extremities. It is defined as a slow growing tumor, which rarely metastasizes but can cause facial deformities if left untreated. A characteristic feature of BCC is the formation of island or nests of basaloid cells found in the epidermis, which can invade the dermis depending on the variant of BCC [36,37]. There are numerous subtypes of BCC; however, several main variants are summarized in Table 2.

## 7. Actinic Keratosis as Precursor Lesions of SCC

Actinic keratosis (AK) is a principal precursor lesion for the formation of SCC. It is often found on parts of the body that are exposed to solar UV radiation such as the forearms, back, scalp, upper chest, face, neck, and back of the hands. Due to the correlation between the development of AK and exposure to UV radiation, it is often found in middle-aged people and the elderly. It can be clinically identified by poorly formed borders, flaky erythroderma, and uneven papules or patches. It is also found on surface areas of the body that exhibit various pre-existing impairments such as uneven pigmentation, telangiectasias, and atrophy. The formation of AK can lead to the development of invasive SCC; however, it can also spontaneously regress or remain a benign AK lesion. Although most cases of SCC have been linked to AK, it has been reported that only 5–10% of AK lesions develop into invasive SCC. Histologically, actinic keratosis is the exponential growth of abnormal keratinocyte cells, which predominately occurs in the lower layers of the epidermis [38]. Additionally, these atypical keratinocytes display an assembly of distinct characteristics such as pleomorphic and hyperchromatic nuclei, polarity deficiency, cell size enlargement, and mitosis enhancement. Several subtypes of AK have been identified that exhibit a broad range of histologic patterns (Table 3) [38,39].

Reports indicate that the rate of progression from AK to SCC ranges from approximately 12% to 20% [40]. A study by Sahin et al. assessed the degree of AK advancement and its link to SCC formation. In this study, evaluations were performed on 115 lesions from 82 patients diagnosed with AK, over a period of eight years, in which the percentage of male patients was 51% and female patients was 48%. This study revealed that the highest percentage of AKs were located on the nose (30.4%), followed by the face (23.5%), lips (8.7%), and ears (7.8%) [41]. Furthermore, when comparing AK localization in males and females, it was observed that AKs frequently occur on the nose of males, whereas in females, they occur on the facial skin. When comparing the subtypes of AK, research indicates that proliferative AK has the highest rate of occurrence at 29.6%. Additionally, acantholytic AKs display a 4% progression rate to SCC [41].

**Table 3 molecules-26-01979-t003:** Various subtypes of actinic keratosis.

Actinic Keratosis Subtype	Characteristics	Occurrence Percentage (%)	References
Pigmented actinic keratosis	Excess quantity of melanin resulting in hyperpigmentation Can be clinically and histologically misdiagnosed as melanoma in situ (accumulation of abnormal elastin) Rough or scaly papule or plaque that is brown or grey (1 to 5 cm diameter) Spreads horizontally across the skin’s surface	1.7	[39,41,42]
Lichenoid actinic keratosis	Dense infiltration of lymphocytes at the dermal–epidermal junction, including basal keratinocyte necrosis Can be morphologically misdiagnosed as benign lichenoid keratosis or lichenoid regression in melanoma Pink to red–dark red scaly plaque on the chest, back and legs	-	[39,43]
Bowenoid actinic keratosis	Atypical keratinocytes inhabit the majority of the epidermis (similar to Bowen’s disease) Does not infiltrate the outer root sheath of the hair follicle Irregularly shaped cells containing light-toned cytoplasms and clustered nuclei, which can develop into large SCCs	9.6	[39,44]
Proliferative actinic keratosis	Flaky, erythematous macules, with indistinct borders Finger-like projections emerge from abnormal keratinocytes that are seen in the superficial dermis Larger than 1 cm and can increase to 3 to 4 cm over time Can expand into the dermis and epidermis; however, however these cells have poor cellular differentiation	29.6	[39,44]
Hypertrophic actinic keratosis	Characterized by increased keratin formation in the stratum corneum and epidermal hypergenesis Histological patterns include focal parakeratosis, abnormally increased thickness of stratum granulosum, amplified epidermal hyperplasia (mimics psoriasis), and dense collagen bundle fibres in the papillary dermis Commonly occurs in the upper extremities of the body	27	[39,45]
Atrophic actinic keratosis	Atrophic transformations present, observable by the decreased epidermis thickness and flattened rete ridges Irregular cells are frequently observed in the basal layer of the epidermis Infrequent mitoses, indicating that this variant emerges from mutations in the basal layer of the epidermis	8.7	[39,46]
Acantholytic actinic keratosis	Acantholysis of atypical keratinocytes, resulting in abnormal keratinocyte separation and intra-epidermal cleft formation Fissures observed within dyskeratotic and acantholytic cells, located in the suprabasal layer Potential to develop into adenoid squamous cell carcinoma	18.3	[39,47]
Actinic cheilitis/cheilosis (rare variant)	Premalignant inflammatory condition that can progress to squamous cell carcinoma Identified by the presence of swollen reddish lesions, which have an excessive amount of fluid (acute phase) Lesions appear grey-whitish, wrinkled, and hyperkeratotic (chronic phase; months–years)	3.5	[48]
Cutaneous horn (uncommon variant)	A hyperkeratotic nodule, which is conical, dense, and projects through the skin Comprised of compacted keratin and often develops on the upper parts of the face Several skin lesions could emerge from the base of this keratin horn	1.7	[49]

## 8. Squamous Cell Carcinoma

### 8.1. Squamous Cell Carcinoma In Situ

Squamous cell carcinoma in situ (SCCIS) can be viewed as a fundamental transitional phase from AK to invasive SCC. Currently, the predominant trend in oncology research is the synonymity that exists between Bowen’s disease and SCCIS, exclusively for lesions occurring on non-genital areas [38,39].

Squamous cell carcinoma in situ, which is also known as Bowen’s disease, can be histologically identified by the following characteristics: atypia, which extends through the complete thickness of the epidermis, excluding the adnexal structures, and hyperparakeratosis, which can either be nominal or extremely abundant, which gives rise to a cutaneous horn. Atypical keratinocytes exhibit apoptosis, hyperchromasia, nuclear pleomorphism, as well as a “windblown” appearance when polarity is lost. It is generally defined as a skin disease that does not possess the ability to invade the dermis layer of the skin. Squamous cell carcinoma in situ can develop on any epidermal body site; however, studies indicate that approximately 72% of SCCIS cases occur on sun-exposed skin surfaces, namely the hands, neck, and head. It is therefore often diagnosed in elderly people aged over 60 years old and is rarely diagnosed in individuals under the age of 30. Other areas in which SCCIS can be found include the nail bed, soles of the feet, and palms of the hand. Squamous cell carcinoma in situ can be clinically distinguished by the following features; presence of a distinct plaque or scaly patch, which is benign and displays erythema. Lesions may develop unfavorable properties which include crustation, fissures, hyperkeratosis, and ulcerations. Reports indicate that the risk of SCCIS progressing to invasive SCC is approximately 3–5%. Statistics further revealed that 20% of tumors that evolve to invasive SCC will ultimately metastasize [38,39].

### 8.2. Invasive Squamous Cell Carcinoma (SCCI)

Invasive squamous cell carcinoma is often referred to as standard SCC. Histologically, it is defined as the vertical invasion of abnormal cells, which originates at the basement membrane and advances into the dermis. It has been reported that approximately 97% of SCCI cases are linked to the malignant progression of AK [50]. There are several variants of SCC (Table 4).

## 9. Recurrent and Metastatic KC

Tumor recurrence and incomplete excision of the primary tumor are major risk factors that can increase the possibility of developing metastatic SCC. Studies have revealed that aggressive tumors are susceptible to recurrence and are accountable for approximately 25–30% of SCC metastasis [60]. In a study by Clayman et al., 130 patients, which exhibited advanced or aggressive SCC tumors, displayed a recurrence of 27.5% [61]. Leading characteristics of recurrent tumors include large tumor size, invasion of lympho-vascular or perineural system, and tumor infiltration into subcutaneous tissue [62]. A study that comprised of 603 patients with cutaneous SCC revealed that 89% of patients died from distant metastasis, which is consistent with current SCC metastatic reports [63]. Key features that are linked to metastasis include cancerous cells that disseminate via the lymphatic, which are responsible for 80% of metastases [64], and the fact that the vast majority of metastatic SCC cases occurred on the head and neck of patients [60]. A report by Lazarus et al. presented a study in which 6900 patients had SCC; however, only 142 patients exhibited metastatic lesions. An evaluation of these lesions revealed that lymph nodes are the primary site for metastases occurrence (Table 5) [65]. On the contrary, metastatic BCC cases are rare, with a percentage rate of approximately 0.0028–0.55% [66]. Metastatic BCC follows the same trend as SCC, where 85% of cases originate from tumors that are located in the head and neck region of the patient. Moreover, statistics reveal that a minimum of two-thirds of metastasis cases arise from tumors that are exclusively situated on the face. Tumors that exhibit a specific set of characteristics can be categorized as having a high metastatic potential, such as tumors located in the mid-face or ear, a tumor that has been present for a long period of time, tumor diameter size (>2 cm), and previous radiation treatment [67]. A study by Freitas et al. evaluated 25 BCC cases between the time period of January 2012 and March 2017 and concluded that metastases occur most frequently in lymph nodes (Table 5).

## 10. Risk Factors Associated with the Development of KC

There are various environmental and biological factors that contribute to the development of KC, which are divided into external and internal factors. Identification of high-risk patients allows for early diagnosis and an efficient treatment regime to be established.

### 10.1. External Risk Factors

#### 10.1.1. Solar UV Radiation

The main external risk factor for developing KC is exposure to solar UV radiation [68]. However, specific patterns of UV radiation exposure result in development of various types of KC. The development of SCC can be linked to long-term sun exposure, whereas BCC formation is associated with excessive sun exposure that transpired in early life as well as intermittent exposure [69]. Furthermore, Kim et al. highlighted that 90% of KC cases are linked to high levels of UV radiation exposure [70]. Solar UV radiation can be divided into three categories, according to a difference in wavelength, namely UVA (320–400 nm), UVB (280–320 nm), and UVC (200–280 nm) [71]. A study conducted by Grossman et al. showed that solar UV radiation induces KC development through DNA damage and immunosuppression [72]. Previously, UVA was reported to primarily be responsible for skin aging; however, recently, UVA has been coupled with UVB, where both are implicated in the development of cutaneous skin cancer. There are different mechanisms by which UVA and UVB cause DNA damage. UVB is known to play a greater role in KC development, due to wavelength penetration depth. UVB radiation is absorbed by cellular components that are present in the epidermis, whereas UVA radiation infiltrates into the basal layer of the epidermis and dermal fibroblasts [73,74]. A study conducted by Boukamp [75], confirmed the findings by Grossman et al. [72] with regards to UVB being a major contributing factor for KC development [75]. This study revealed that the formation of photoproducts, which damage the DNA present in keratinocytes, occurs due to long-term UVB exposure [75]. One of the major photoproducts that form is cyclobutene pyrimidine dimer (CPD). CPDs form thymine dimers (T/T) and pyrimidine-pyrimidone lesions (6-4PP), which, if unrepaired, are mutagenic. This specific type of DNA damage can be repaired by the nucleotide excision repair mechanism (NER); however, malfunction of this mechanism can result in multifocal skin cancer [76]. Additionally, UV radiation has been reported to cause mutations in the suppressor gene existing in the p53 protein. The central function of the p53 tumor suppressor gene is to encode for a protein, which regulates the cell cycle and induces apoptosis [76]. Previous studies have indicated that mutations present in the p53 gene are associated with numerous human cancers and are most prevalent in KCs [77]. According to Kooy et al., UV radiation is able to induce immunosuppression by depleting epidermal dendritic Langerhans cells (CD1a+). As a result, T helper type-1 converts to T helper type-2 response, which inhibits the ability of cell-containing-antigens to induce antitumor immunity [78].

#### 10.1.2. Indoor Tanning

Artificial sources of UV radiation have been categorized as cancer causing agents by the International Agency for Research on Cancer (IARC) in the year 2012 [79]. Indoor tanning generates similar adverse effects in human skin as exposure to solar UVB radiation; however, reports indicate that indoor tanning is 10–15 times stronger than exposure to solar UVA radiation [80]. A review conducted by Wehner et al. investigated approximately 9300 cases of keratinocyte carcinoma and was able to correlate the development of KC to indoor tanning. This report revealed a 67% higher risk of developing SCC and a 29% higher risk for BCC development when exposed to indoor tanning. Studies indicate that exposure to indoor tanning between 16–25 years leads to a greater risk of developing BCC [81].

#### 10.1.3. Ionizing Radiation

The association between exposure to ionizing radiation and skin cancer formation was discovered by radiologists that were working with X-ray analysis. Furthermore, SCC development was observed in skin locations that exhibited dermatitis and ulceration, which was caused due to high-level ionizing radiation exposure, whereas formation of BCC was seen in low to moderate exposure to ionizing radiation [82]. Ionizing radiation is used as a form of treatment for various types of cancers; however, this is often associated with the development of radiation dermatitis, which occurs in approximately 95% of patients receiving radiation therapy. This is due to damage caused to the basal keratinocytes and hair follicle stem cells, which is followed by double-stranded DNA breaks and inflammation caused by the production of reactive oxygen species [83].

#### 10.1.4. Arsenic Exposure

Studies indicate that the two leading characteristics that arsenic exhibits are its ability to act as a toxin and as a carcinogen. The mechanism of action is to target and negatively alter the cellular processes within various organ systems based on a dose and time-dependent manner. The toxic effects of arsenic are first displayed in the skin, which can include the development of BCC and SCC. Development of SCC from arsenic exposure is said to be more aggressive in nature when compared to chronic UV-induced SCC. Statistics reveal that 33% of untreated arsenic-induced SCC demonstrates metastatic behavior [84].

### 10.2. Internal Factors

#### 10.2.1. Age

Studies have shown that the development of KC is more prevalent in elderly patients with a median age diagnosis of 70 years and older [85]. There are several factors that potentially contribute to the frequent occurrence of KC in the elderly, namely prolonged exposure to solar UV radiation and repair mechanisms of the cell that are not functioning at optimum levels [86]. Transformations that occur in the immune system of elderly patients result in immune suppression, which can lead to opportunistic disease development, of which malignancies are frequently observed [87].

#### 10.2.2. Skin Type

Skin type and pigmentation is a crucial factor regarding the development of KC. Skin type 1 is represented by people who have fair skin, light blue, and grey eyes, with light red and blonde hair, and are more susceptible to KC formation. The number of KC cases occur more frequently in fair-skinned individuals as compared to dark-skinned individuals. This is due to low levels of melanin present in the skin. Melanin, which is responsible for skin pigmentation, has photoprotective properties. It protects the skin by acting as a physical barrier to UV radiation and is furthermore able to absorb UV radiation, thereby limiting the amount of UV that penetrates the skin. Melanin is produced in the melanosomes, which is translocated to adjacent keratinocytes via dendrites [88,89].

#### 10.2.3. Immunosuppression

Immunosuppression is a significant contributing factor for the development of KC, more specifically SCC. The development of KC occurs more frequently in patients that exhibit immunodeficiency, which includes patients that use immunosuppressive drug therapy in cases such as solid-organ transplantation, auto-immune inflammatory diseases, and human immunodeficiency virus (HIV) infection. Patients who have received organ transplants have an increased risk (20–200-fold) of developing SCC. A study revealed the incidence ratio of SCC caused by solid-organ transplantation to be 1355/100,000, whereas the incidence ratio of SCC in the general population was 38/100,000 [90].

## 11. Topical Pharmacotherapies Currently Used for the Treatment of KC

Topical chemotherapy is a form of non-surgical therapy used to remove or eliminate localized skin cancer cells. There are several topical treatments available (Table 6), which can induce cell death through the direct damage to DNA/RNA, such as 5-fluorouracil, or act as immune-modulators, such as imiquimod, which stimulates the production of various cytokines, thereby inducing antitumor activity. Topical treatments are often a consideration when patients are elderly or unhealthy; therefore, surgery may not be an option, or for individuals who have tumors/lesions on areas that cosmetically sensitive, and therefore surgery may result in a disfiguring scar [91].

## 12. Transdermal Delivery of Drugs

A fundamental approach used to enhance bioavailability of pharmaceutical drugs is developing novel drug delivery systems. Although oral delivery systems are preferred for pharmaceutical drug administration, these systems are linked to several disadvantages such as insufficient drug stability within the gastrointestinal tract, reduced drug concentration upon reaching its site of action due to metabolism, and decreased drug solubility in intestinal fluid resulting in poor permeability through the intestinal membrane [117]. Due to the large surface area and accessibility of the skin, extensive investigations of drug delivery via the skin have been conducted. Administration of drugs through the skin can result in the drug being confined within the skin (topical application) or the drug can penetrate through the skin, allowing it to reach the blood circulatory system (transdermal delivery). There are numerous advantages associated with transdermal drug delivery when compared to conventional drug administration routes, these include; non-invasiveness, increased patient compliance, enhanced drug bioavailability, more cost effective and decreased drug plasma fluctuations [118]. However, it is imperative to consider physiological and physiochemical factors, as these factors influence a drug’s movement through skin. A drug’s absorption rate is primarily controlled by skin age, moisture contents, and anatomical location. Aged skin has a decreased moisture contents, resulting in reduced and slower absorption when compared to younger skin. An increase in temperature leads to an enhanced absorption rate, whereas a reduction in blood flow results in a drug’s influx being negatively impacted [119]. An ideal transdermal drug should possess characteristics such as a short half-life, low molecular weight (less than 1000 Da), bio-compatible, low melting point, and an affinity for both hydrophilic and lipophilic phases [120].

### 12.1. Targeted Delivery Through the Skin

The main objective of drugs that exert their pharmaceutical effect topically is to target a multitude of sites that exist in different skin layers, skin appendages, and underlying tissue. Studies indicate that the systemic circulatory system is predominately targeted by transdermal drug compounds; moreover, the anatomical structures of interest are hair follicles, nerves, Langerhans cells, keratinocytes, and melanocytes within the epidermis. It is imperative that drug candidates for transdermal delivery include the following characteristics: limited sites for hydrogen bonding, have a low molecular weight, average lipophilicity, and a low melting point [121]. The permeation of a drug occurs through the stratum corneum and can be calculated by Fick’s second law:(1)$$J=\;\;\frac{DmCvP}{L}$$
where *J* represents the transport flux, *Dm* is the diffusion coefficient of the drug present in the membrane, *Cv* represents the drug concentration that exists in the vehicle, *P* is the drug partition coefficient, and *L* represents the thickness of the stratum corneum [122].

### 12.2. Transdermal Drug Permeation Routes

Administration of drugs through the skin can follow two potential routes, trans-epidermal and trans-appendegeal (Figure 2) [117]. The trans-epidermal pathway allows for molecules to pass through the stratum corneum and can be further sub-divided into intracellular and intercellular pathways [123]. The intracellular route depicts a pathway that permits the transport of hydrophilic or polar solutes through differentiated keratinocytes known as corneocytes, whereas the intercellular route allows the transport of lipophilic or non-polar solutes via intercellular spaces in the lipid matrix. The second transdermal drug route (trans-appendageal route) represents a pathway that allows the permeation of molecules via hair follicles that are associated with sebaceous glands as well as sweat glands [123]. This route is suitable for ions and large polar molecules, which experience difficulty in stratum corneum permeation [124].

### 12.3. Transdermal Delivery of Skin Cancer Drugs

The majority of chemotherapeutic drugs are distributed using the systemic circulatory system and are capable of generating cytotoxic effects on healthy cells. Therefore, transdermal drug delivery of anticancer compounds can be viewed as an attractive alternative, due to the two essential advantages of transdermal drug delivery systems, namely increased therapeutic benefit and enhanced drug targeting. However, there are several challenges that have been linked to this treatment method, including increased concentration, lack of bioavailability, and penetration of drug compounds possessing antineoplastic properties [125,126]. Due to the advancements in technology, anticancer macromolecules, inclusive of proteins and nucleic acids, that experience difficulty penetrating the stratum corneum can be delivered with the assistance of penetration enhancers, physical enhancement devices, and micro-carriers [127].

Studies have reported on two types of penetration enhancers, namely chemical and biological. Chemical penetration enhancers can be described as compounds that facilitate increased drug penetration through the skin, such as alcohol, terpenes, esters, fatty acids, polyols, and surfactants. Reports on various mechanisms of action for this type of enhancer include facilitating disturbances within the stratum corneum, intercellular protein interaction, and enhanced drug segmentation within the stratum corneum [128]. Biological enhancers are peptide-based and are capable of transporting an array of compounds, such as nucleic acids, proteins, polymers, and nanoparticles, throughout the skin with minimal toxicity. Physical enhancement devices can be characterized by the implementation of electric fields such as electroporation, iontophoresis, and sonophoresis [127]. Further literature explained that the fundamental purpose of these techniques is based on the ability to exert a momentary and reversible disintegration of the stratum corneum, which results in enhanced permeation of an antineoplastic drug at a tumor site [129].

In recent times, micro-nanocarrier systems that include inorganic nanoparticles, nano-emulsions, dendritic nanocarriers, and liposomes have attracted increased attention in the field of transdermal drug delivery, as they offer various advantages, such as increased skin penetration, enhanced solubility, and implementation of controlled release [127]. Within the micro-nanocarrier system, there exist several elements that greatly contribute to skin permeability, such as charge, shape, and size of nanomaterials [130]. 

This review explored several transdermal drug delivery and micro-nanocarrier systems for their application in skin disorders and skin cancer (Table 7).

## 13. Potential Therapeutic Effects of Phytochemical/Medicinal Plants against KC

Due to the considerable incline of KC cases worldwide, researchers have a vested interest in developing novel treatment options [139]. The fundamental purpose of an anticancer therapeutic strategy is the ability to selectivity target malignant tumor cells, that either inhibits their growth or induces cell death [140]. However, the vast majority of KC treatments that are currently available, also have a detrimental effect on non-tumors cells. In an effort to create cutting-edge, more effective and non-toxic anticancer treatments, researchers have explored the possibility of isolating compounds from natural sources, more specifically phytochemicals [141].

Studies have reported that phytochemicals and medicinal plants exhibit potential anticancer properties. From the year 1940 to 2014, approximately 50% of approved anticancer therapeutic agents were either obtained or derived from natural sources [142]. Several medicinal plants, and their bioactive compounds, have been investigated for potential anticancer activity. These have been tested against skin tumors, in both in vitro and in vivo studies, and have exhibited noteworthy activity by arresting the development and proliferation of skin tumor cells [143]. Phytochemicals have the potential to alter various molecular processes associated with the development of skin cancer and subsequently inhibit tumor proliferation [144]. Some of the reported studies on medicinal plants and phytochemicals have been summarized in Table 8. Additionally, emerging KC treatments that are currently in the clinical trial phase have been summarized in Table 9.

Often, public perception is that herbal or medicinal plant treatments are deemed safe and effective, as is the case with the use of black salve. Black salve has not undergone any clinical trials in order to evaluate its safety or efficacy against skin cancer and is available to purchase over-the-counter or online. A review by Lim summarizes numerous case reports on the use of black salve for skin neoplasms; however, almost each of the cases reported side effects that included severe tissue necrosis and damage as well as scarring ulceration [176]. A major alkaloid compound present within bloodroot is sanguinarine, which has been linked to the potential carcinogenic effect of bloodroot; however, due to contradictory in vitro and in vivo results, the carcinogenic classification of sanguinarine has not yet been established [177]. 

Furthermore, there is a lack of quality control and quantification or standardization of the active ingredients within these “home” remedies. Black salve has been largely linked to the occurrence of facial deformities, which needs to be corrected through cosmetic reconstruction, which emphasizes the need for the correct safety and efficacy trials to be conducted before these types of remedies are made available on the market for use. 

Similarly, in a pilot trial, betulin oleogel-S10 was initially shown to be effective in treating AK, with a clearance rate of 64%; however, this was a small trial and did not include the use of a placebo control [178]. Consequently, a larger study on the use of betulin oleogel-S10 was conducted by Pflugfelder et al. [175], including a placebo control group, which concluded that the topical use of betulin-oleaogel-S10 did not significantly clear AK when compared to the placebo group. 

Camptothecin, an alkaloid isolated from *Camptotheca acuminata* Decne. is a cytotoxic compound that inhibits DNA topoisomerase I [179]; however, due to its insolubility in biocompatible solvents, it remains a challenge to effectively administer it intravenously. However, in a study by Lin et al. [132], α-MSH liposomes that encapsulate camptothecin showed an enhanced antiproliferative activity against melanoma cells, when compared to camptothecin alone, due to the targeted delivery and controlled release of camptothecin, which emphasizes the importance of using appropriate drug delivery systems. Similarly, a study by Kessels et al. [171] showed that a 10% sinecatechin ointment, which contained EGCG, was not effective in treating superficial BCC; however, it was concluded that this may be due to a lack of EGCG uptake by the cancerous cells and that using liposomes as a drug delivery mechanism may enhance the uptake and effectiveness of EGCG.

## 14. Conclusions

The number of KC cases is increasing at an alarming rate globally. Moreover, statistics revealed that South Africa is one of the leading countries in the world, with the highest rates of skin cancer cases. Although the current topical treatments available for KC have shown promising results, they have also been associated with numerous adverse effects. Therefore, the discovery, development, and treatment of KC using plants and their natural products are becoming increasingly sort after. Natural plant compounds, such as polyphenolics and alkaloids, have in numerous studies demonstrated increased antiproliferative and anticancer activity. Studies have shown that these natural compounds are capable of functioning both independently and interdependently. Various reports further indicated the emergence of nanocarrier and transdermal delivery systems and the numerous advantages they offer, specifically to overcome current challenges faced in cancer treatment, such as bioavailability, targeted delivery, and systemic toxicity. However, it is important to note that case studies alone are not considered conclusive to determine or evaluate the safety and efficacy of a test substance such as a plant extract or natural product. Large clinical trials are required that include a placebo control group and different skin types in order to determine the effectiveness of a test substance.

Although there have been numerous reports on the antiproliferative activity of plant extracts and their natural products on non-melanoma skin cancer cell lines, there have been relatively few studies that have evaluated the potential of these plants/natural products in clinical trials. Additionally, although some plant extracts or natural products may not have shown significant in vitro activity or anticancer activity in animal models when used alone, a suitable drug delivery system may provide for enhanced delivery and efficacy and therefore should further be explored. 

The topical application of EGCG and caffeine, using different transdermal delivery systems, should be considered for further evaluation, as this may result in increased uptake of phytochemicals by the cancerous cells. In addition, the use of capsaicin against skin cancer should be further assessed, as it showed promising results in two patients but has not been assessed in larger clinical trials. The potential of synthesizing capsaicin derivatives may provide a source of novel compounds with decreased irritant effects and increased efficacy. Ursolic acid, which has extensively been studied for its activity against melanoma, has not been well documented for its activity against keratinocyte carcinomas and therefore should be considered for further assessment. 

It is, however, important to note that upon identification of a plant extract or compound showing significant activity, the mechanism of action, as well as the efficacy and toxicity potential of the identified extract or compounds requires further evaluation using clinical trials, which includes the pharmacodynamics and pharmacokinetic properties of the plant extract/compound.

## Figures and Tables

**Figure 1 molecules-26-01979-f001:**
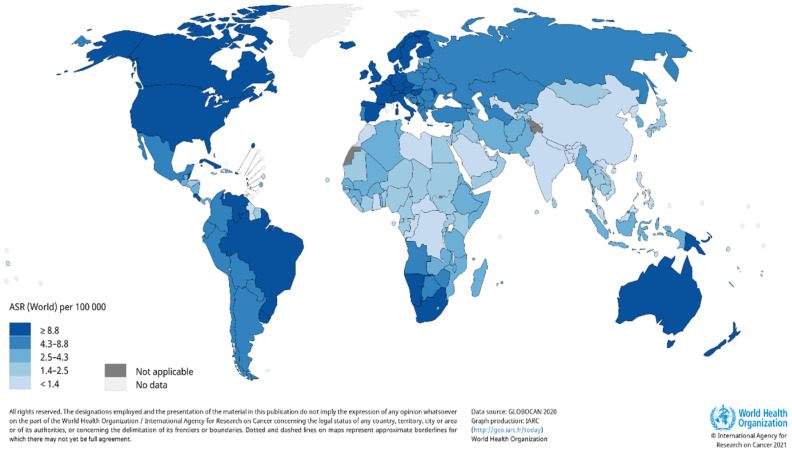
Estimated age-standardized world-wide incidence rates of non-melanoma skin cancer in 2020. Reproduced from [22]. Copyright 2021, International Agency for Research on Cancer 2021, World Health Organization.

**Figure 2 molecules-26-01979-f002:**
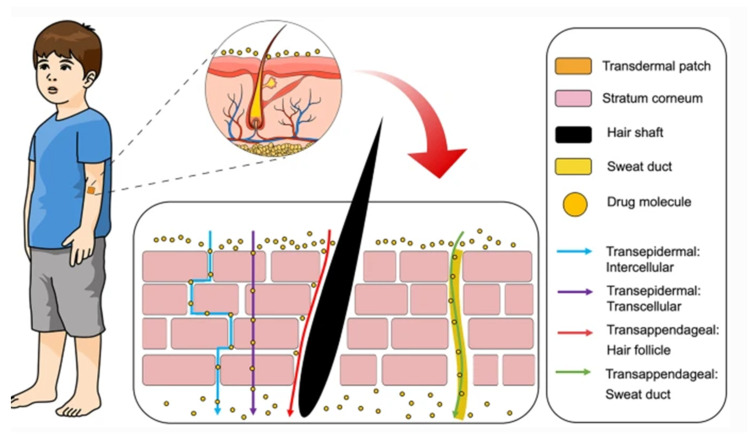
Transdermal delivery pathways in the skin. Reproduced with permission from [117]. This is an open access article distributed under the terms of the Creative Commons CC BY license.

**Table 1 molecules-26-01979-t001:** The number staging system of keratinocyte carcinoma [34].

Stage	Description
0	In situ carcinoma, cancer has developed but has not spread or grown into surrounding tissue
1	Tumor ≤2 cm in size, with less than two high-risk features
2	Tumor >2 cm in size, with two or more high-risk features
3	Tumor with invasion of the maxilla, mandible, orbit, or temporal bone or the tumor has spread to nearby lymph nodes (<3 cm in size)
4	Tumor with invasion of skeleton or perineural invasion of skull base or the tumor has spread to lymph nodes (>3 cm in size) or an internal organ

High-risk features include >2 mm thick, cells are poorly differentiated or undifferentiated, has grown into the dermis, perineural invasion (space around a nerve), primary site non-hair-bearing lip, started to develop on the ear or lip.

**Table 2 molecules-26-01979-t002:** Clinical variants of basal cell carcinoma.

Basal Cell Carcinoma Variant	Characteristics	References
Nodular BCC	Most dominant form of BCC (60–80% of BCC cases) Most often arises on sun exposed areas, commonly on the head and neck (85–90%) Develops over a long period of time, often years Identified by elevated, exophytic pearl-shaped nodules with telangiectasia on the surface and periphery Bleeding and ulceration often occurs Subtype: micronodular BCC, less susceptible to ulceration, and is skin or great in color with a firm shape and defined border	[38,39,40,41]
Pigmented BCC	Rare variant of BCC, consists of 6% of BCC cases Pigmented variant of nodular BCC (can be found in micronodular and superficial BCC) Brown, black, or blue in color with enlarged pigmented papule and telangiectasis Often misdiagnosed as angiomas, seborrheic keratisis, or melanoma	[39,40,42]
Superficial BCC	Consists of 10–30% of BCC cases, often affecting younger patients Slow-growing and develops over a long period of time, often on the trunk or upper extremities Flat, glazed, pale pink lesion with distinct borders, enveloped marginally with protruding edges	[39]
Morphoeic/sclerosing BCC	Aggressive type of BCC, occurring on the face Can be fast-growing, reaching several centimetres in a few months, or undergoes no changes for several years Slightly glistening surface with indistinct boarders	[39]

**Table 4 molecules-26-01979-t004:** Clinical variants of squamous cell carcinoma.

Squamous Cell Carcinoma Subtype	Characteristics	References
Generic/simplex SCC	Atypical keratinocytes present in the form of lobules and cords Originates in the epidermis and extend to the dermis Forms small/large islands or invasive tumor strands Characteristics include a mononuclear inflammatory infiltrate, loss of surface epithelial cells, ulceration, and hyperkeratosis	[51]
Acantholytic SCC (adenoid/lobular SCC)	A form of sweat gland carcinoma Identified by squamous differentiation related to acantholysis (forms forged glandular tumors) High-risk variant of SCC Forms clefts within the tumors due to loss of cohesion between cells	[52,53]
Spindle cell SCC (sarcomatoid SCC)	A rare variant of SCC that is not well differentiated Infiltration of proliferating pleomorphic cells in the connective tissue Ability to invade the dermis, subcutis, fascia, muscle, and bone Atypical cells can developed into a whorled pattern, with the ability to invade the dermis Presence of extremely large cells that are multinucleated, pleomorphic, and have multiple mitotic structures Develops on sun-exposed areas such as the head, neck, chest, and upper extremities	[38,54]
Verrucous SCC	Well-differentiated SCC comprising of rete ridges (bulbous, thickened, and papillomatous), which invade the dermis Four categories (depending on where it develops): oroaerodigestive VC, anourogenital VC, palmplantar VC, and cutaneous VC Tumor strands form sinus tracts (which attached to the skin’s surface) that invade the dermis and subcutaneous layer Presence of mitotic structures, nuclear growth, large atypical keratinocytes, and hyperchromasia Often associated with the human papilloma virus	[38,55]
Clear-cell SCC (hydropic SCC)	Rare variant of SCC that appears edematous with ulcerated masses or nodules Three types: Type 1 (keratinizing), Type 2 (non-keratinizing), and Type 3 (pleomorphic) Type 1: vacant cytoplas, lesions appear as a sheet formation, or tumor cells appear as small clusters sporadically dispersed Type 2: emerges from dermis and tumor cells are in a parallel formation (separated by stroma, which is fibrotic and inflammatory in nature) Type 3: originates in the epidermis and exhibits severe ulceration	[56]
Single cell infiltrates	Rare variant of SCC, identified by the presence of single infiltrates Often occurs in the elderly found on the face and neck More aggressive in nature when compared to generic SCC Lesions are often undetected or misdiagnosed Irregular cells are individually arranged or clustered in the dermis Located in the skin where there is excess elastin due to sun damage	[57]
*De Novo* SCC	Aggressive variant of SCC not associated with actinic keratosis or sun exposure Well-differentiated simplex SCC, in close proximity to an ulcer or scar Develops on areas of the skin exposed to long-term disease or injury, commonly in lower extremities	[58,59]

**Table 5 molecules-26-01979-t005:** Anatomical sites of metastases occurrence in squamous cell carcinoma and basal cell carcinoma.

Site of Metastases	Occurrence Percentage (%)
**Squamous Cell Carcinoma**	
Lymph node	4.3
Lung	0.2
Liver	1.1
Bone	0.2
Subcutaneous tissue	0.2
Brain	0.2
Generalized	0.1
Site unspecified	1.6
**Basal Cell Carcinoma**	
Lymph nodes	56
Lung	36
Paotid gland	20
Bone	16
Submandibular gland	12
Thyroid	4
Skin	0
Liver	0

**Table 6 molecules-26-01979-t006:** Existing topical pharmacotherapies for the treatment of KC.

Pharmacotherapy	Efficacy	Mechanism	Disadvantages/Adverse Side Effects	References
5-Fluorouracil (5-FU)	5% FU: 80% and 54–86% efficacy for superficial BCC and SCC in situ, respectively 30 mg/mL (one-thrice weekly) intralesional 5-FU: 90–100% efficacy for small superficial/modular tumors (0.6–1.5 cm) 50 mg/mL (biweekly) intralesional 5-FU: 67% efficacy for large tumors (2.4 cm)	Disrupts DNA synthesis and repair by inhibiting thymidylate synthase; causes DNA damage, DNA strand breaks and cell death Misincorporation of 5-FU in RNA; inhibits conversion of pre-rRNA to mature rRNA and disrupts post-transcriptional modification of tRNA	High rate of tumor recurrence; optimal for small-sized tumors Erythema, scaling, blisters, necrosis, ulceration, erosions, pruritus, burning, headaches, fever, diarrhea, nausea, and mouth ulcers	[92,93,94,95,96,97]
Imiquimod (IMQ)	5% IMQ: 43–94% for superficial BCC 5% IMQ: 50–65% for nodular BCC 5% IMQ: 71% for invasive SCC 5% IMQ: 57–80% for Bowen’s disease	Induces pro-inflammatory cytokines secretion, interferon gamma (IFN-γ), tumor necrosis factor alpha (TNF-α), interferon alpha (IFN-α), interleukin (IL)-6, IL-1α, IL-1β, IL-8, and IL-12, thereby activation acquired and natural immune response and antitumor activity	Extensive recurrence rate after first 12–24 months; optimal for tumors <2 cm in size Erythema, discomfort, erosion, scaling, blisters, necrosis, ulcerations, erosions, pruritus, burning, flu-like symptoms, dizziness, and headaches	[95,96,98]
Ingenol mebutate (IM) Active compound in *Euphorbia peplus* L. (milkweed) sap	0.05% IM: 63% efficacy for superficial BCC (increased efficacy directly proportional to higher dosage) 0.25% IM gel: 70 and 30% efficacy for SCC growth in female and male mice respectively	Induces cell necrosis through loss of mitochondrial membrane potential and induction of mitochondrial membrane polarization Necrosis induced pro-inflammatory cytokines resulting in neutrophil-mediated antibody-dependent cellular cytotoxicity	Thicker skin in male mice resulted in poorer drug penetration and decreased efficacy Crusting, flaking, erythema, erosion/ulceration, swelling, and blistering	[99,100,101,102,103]
Photodynamic therapy (PDT)	72–100% efficacy on superficial BCC	Increased uptake of PDT by cancerous cells; once PDT has exited normal cells, tumor cells (with PDT) are exposed to light at a specific wavelength, resulting in release of reactive oxygen species, thereby inducing cell death	Nodular BCC and BCC tumors >2 mm are less responsive to PDT (inadequate penetration); BCC tumors between 1–2 mm in thickness, SCC an AK lesions have a high recurrence rate; less effective on superficial BCC than IMQ and 5-FU Burning, prickling, erythema, edema, hypo-and hyper-pigmentation, allergic contact dermatitis (rare) and pain, which often leads to incomplete treatments	[104,105,106,107,108,109,110,111,112,113]
Retinoids Class of compounds derived from vitamin A	0.1% tazarotene gel (daily for eight months): cleared 11 of 13 superficial BCC and 5 of 17 nodular BCC; a 24 week trial recorded 70.8% of BCC with >50% regression and 30.5% healed with no recurrence after 3 years 0.1% tazarotene gel (daily): efficacy of 46.6% of SCC in situ (0.5–4 cm); complete clearance from month 3–5 (no recurrence after 3 month follow-up)	Antiproliferative activity and induction of apoptosis in basaliomatous cells	Effective against undifferentiated BCC tumors, however not effective against keratotic BCCs (overexpression of p53 and cellular retinol binding protein-1) Mild erythema, edema, and local skin irritation	[114,115,116]

**Table 7 molecules-26-01979-t007:** Transdermal drug delivery and micro-nanocarrier systems in skin disorders and skin cancer.

Nano-Carrier	Study	Outcome	Reference
Liposomes	Synthesized elastic liposomes loaded with 5-fluorouracil (5-FU) investigated (in vitro and in vivo) for drug permeation enhancement across the stratum corneum of the skin.	Optimized elastic 5-FU loaded liposomes showed higher drug permeation flux (89.74 ± 8.5 μg/cm^2^/h^2^) when compared with the drug solution 5-FU (8.958 ± 6.9 μg/cm^2^/h^2^) and the liposome (36.80 ± 6.4 µg/cm^2^/h^2^) alone Drug deposition of the optimized elastic 5-FU loaded liposomes was approximately three-fold higher in comparison with the 5-FU drug solution In vivo analysis showed that the optimized elastic 5-FU loaded liposomes enhanced drug permeation without generating skin structure transformation	[131]
	Uptake of α-melanocyte-stimulating hormone (α-MSH)-conjugated liposomes in melanoma cells (B16-F10)	Increased uptake in melanoma cells when compared to conventional liposomes Camptothecin encapsulated by α-MSH-conjugated liposomes resulted in a sustained and controlled release of camptothecin	[132]
	Cytotoxicity of co-delivered curcumin encapsulated cationic liposomes complexed with STAT3 siRNA against SCC cells	Significant reduction in SCC cell growth when compared to the treatment of cells with curcumin and STAT3 siRNA alone	[133]
Solid lipid nanoparticles (SLNs)	Cytotoxicity of doxorubicin-loaded solid lipid nanoparticles against B16-F10 cells and melanoma-induced Balb/C mice	Increased cytotoxicity against B16F10 cells and melanoma-induced Balb/C mice when compared to doxorubicin alone	[134]
	5-FU loaded SLNs for the treatment of skin carcinoma in vivo	Higher permeation of 5-FU loaded SLNs (269.37 ± 10.92 μg/cm^2^) in comparison with the drug solution 5-FU (122 ± 3.09 μg/cm^2^) Mice administered with 5-FU loaded SLNs demonstrated a decrease in angiogenesis, a decline in inflammatory reactions, and reduced keratosis	[135]
Microneedles	Treatment of BCC using the intradermal delivery of an immunomodulator (5% *w*/*v* imiquimod cream), using an oscillating microneedle device (Dermapen). Dermal permeation analysis was performed on the cross-sections of porcine skin	Significant increase in transdermal permeation of 5% *w*/*v* imiquimod cream when the cream was first applied to the skin, followed by the Dermapen application Limited dermal permeation observed with the application of 5% *w*/*v* imiquimod cream alone The enhanced dermal permeation was due to an intradermal depot that was generated, which lasted for 24 h	[136]
Hydrogel	Investigation of injectable intra-tumoral 5-FU hydrogel to enhance efficacy and decrease systemic toxicity associated with 5-FU observed in cancer patients	A single injection of 5-FU loaded hydrogel exhibited enhanced tumor growth suppression when compared with the drug solution or hydrogel alone 5-FU loaded hydrogel showed a longer retention time (>18 days) within the tumors Low biodistribution of 5-FU into other organs was maintained	[137]
Ethosomes	A complex of CUR-Eth-PEI/DOX-Eth-SC (cytotoxic drug and a chemosensitizer) was evaluated (in vitro and in vivo) for potential anticancer activity on B16-F10 cells. Two modified ethosomes were synthesized, namely polyethyleneimine (PEI)-modified ethosomes (Eth-PEI) and sodium cholate (SC)-modified ethosomes (Eth-SC). These modified ethosomes functioned as carriers for doxorubicin (DOX) and curcumin (CUR)	CUR-Eth-PEI/DOX-Eth-SC with a ratio of (7:3) exhibited enhanced antitumor activity in the treatment of melanoma	[138]

**Table 8 molecules-26-01979-t008:** The therapeutic effect of medicinal plants and phytochemicals against keratinocyte carcinoma.

Phytochemical	Source/Origin	Treatment	Outcome	References
Hypericin	*Hypericum perforatum* L. (St John’s Wort)	Hypericin directly injected into affected tissue, 3–5 times weekly (SCC (40–100 μg) and BCC (40–200 μg)), showed no necrosis of surrounding tissue and was successful as a targeted delivery system	Combination of hypericin and PDT resulted in pain and burning	[145,146]
		Effect of 0.07% hypericin on BCC, AK, and Bowen’s disease, followed by irradiation (weekly, for 6 weeks)	All patients experiences pain and burning after irradiation, 50% complete clinical remission of AKs, 11% histological clearance of sBCC, and 80% histological clearance of Bowen’s disease	[147]
		Mice injected with SCC cells to develop tumors (3–15 mm diameter) were injected with 10 µL of DMSO containing 10 µg hypericin per gram of tumor and irradiated after 24 h	Hypericin retained in tumors for a prolonged period of time was observed to be more effective in small sized tumors (<400 mm^3^), whereas larger tumor displayed partial ablation followed by recurrence	[148]
Black salve (escharotic agent) *Sanguinaria canadensis* L. (bloodroot)		Ointment containing 300 mg bloodroot, galangal, sheep sorrel, and red clover resolved suspected melanoma neoplasm of the left naris (63 year old male)	Complete loss of the left naris and severe tissue damage	[149]
		Application on BCC located on the nasal cavity (83 year old male)	Complete loss of nasal ala	[150]
		Application of black and “yellow” salve on micronodular BCC located on right nasal sidewall (65 year old female)	Patient discontinued use due to pain and tenderness, formation of 12 mm ulceration with eschar formation. Secondary intention healing treated the wound	[151]
		Application on 5 mm BCC lesion (51 year old male)	Agonizing pain and formation of large eschar and formation of scar. Biopsy after 12-months showed no presence of BCC	[152]
		Application on BCC located in the right-hand side of the neck (49 year old male)	Development of triangular keloidal scar that had to be surgically removed and repaired. After reconstruction, no tumor was identified	[153]
		Application to SCC on the right lower leg (55 year old woman)	Formation of a thick escharotic plaque, which dislodged revealing normal granulation tissue. Histological examination revealed no residual SCC	[154]
		Application of black salve (containing zinc chloride) on BCC on the left cheek of the face	Formation of a thick escharotic plaque, which dislodged revealing normal granulation tissue. Histological examination of the plaque revealed acute and subacute inflammation, necrosis and BCC in the dermis; however, scar did not have any BCC	[154]
Ursolic acid- UA (pentacyclic triterpene)	Several plant species such as *Ocimum basilicum* L. (basil), *Salvia Rosmarinus* Schleid. (rosemary), apple peels (*Malus pumila* Mill.), and berries	Effect against B16 mouse melanoma cells after 24 h exposure	UA showed a fifty percent inhibitory concentration of 7.7 μΜ. Cytotoxicity attributed to potential inhibition of lipoxygenase and cyclooxygenase and cytostatic activity.	[155]
		Effect on Ca3/7 (mouse SCC) and MT1/2 (mouse skin papilloma) skin cancer cells	Induced cell death in both cell lines through activation of AMP-activated protein kinase (AMPK) and peroxisome proliferator activated receptor-α (PPAR-α)	[156]
Luteolin (flavonoid)	Several plant species such as *Daucus carota* L. (carrots), *Capsicum annuum* L. (peppers), *Petroselinum crispum* (Mill.) Fuss (parsley), and *Brassica oleraea* L. (broccoli)	Effect on B16F10 murine melanoma cells	Inhibited tumor progression by inhibiting hypoxia-induced epithelial-mesenchymal transition in melanoma cells through upregulation of β3 integrin	[157]
		Effect on B16 murine melanoma cells	Induced apoptosis in melanoma cells through ERK1/2 signaling attenuation, upregulation of Bax, and down-regulation of Bcl-2	[158,159]
		Effect in normal human keratinocytes (NHK) after exposure to UVB radiation	Enhanced survival rate of NHK through inhibition of the mitochondrial intrinsic apoptotic pathway and inhibition of inflammatory mediators IL-1α and prostaglandin-E2. However, did not inhibit malignant keratinocytes	[160]
Resveratrol-RV (polyphenol)	Commonly found in *Vitis vinifera* L. (grapes), *Morus* spp (mulberries), and *Arachis hypogaeae* L. (peanuts)	Photo-chemopreventive activity of RV (25 μmol/0.2 mL acetone per mouse) in hairless mice induced with UVB radiation Topical application in hairless mice induced with UVB radiation	Inhibition of skin thickness growth and ear punch weight Topical application inhibited increased ornithine decarboxylase (ODC) enzyme activity and protein expression. Increased levels of ODC activity are linked to an increase in neoplastic growth	[161]
		Effect of RV (1–50 μΜ for 24 h) on human epidermoid carcinoma (A431) cells	Inhibited cell growth, induced apoptosis, and cell cycle arrest at the G1 phase through the activation of the cyclin-dependent kinase inhibitor 1 (WAF1/p21), which in turn induced cyclin D1/D2-cdk6, cyclin D1/D2-cdk4, and cyclin E-cdk2 complex inhibition	[162]
Capsaicin (capsaicinoid)	Major compound present in plants belonging to the *Capsicum* genus	Effect on parental SCC cells	Induced apoptosis due to mitochondrial respiration suppression, antiproliferative activity potentially due to production of hydroperoxide and/or the inhibition of enzymatic processes within the electron transport chain	[163]
		Topical application of alcoholic *Capsicum* extract (containing capsaicin) on BCC and SCC lesions	Topical application reduced the size of the lesion and the lesions disappeared after a certain period of time	[164]
Ethanolic fruit extract of *Combretum molle*		Effect of extract against A431 cells	Antiproliferative activity with IC_50_ value of 23.2 ± 0.8 μg/mL	[165]
Methanolic leaf extract of *Calystegia sepium*			Antiproliferative activity with IC_50_ value of 24.71 μg/mL; induced cell cycle arrest at G0/G1 stage and induced the expression of nuclear factor kappa B1 (NF-κβ) and apoptotic peptidase activating factor 1 (APAF1)	[166]
Ethanolic aerial part extract of *Euclea crispa* subsp. *crispa*			Antiproliferative activity with IC_50_ value of 41.8 ± 0.4 μg/mL	[165]
Ethanolic leaf and stem extract of *Helichrysum odoratissimum*			Antiproliferative activity with IC_50_ value of 15.5 ± 0.2 μg/mL; induced apoptosis and increased IL-12 and inhibited IL-8 levels in U937 cells	[167]
Ethanolic aerial part extract of *Sideroxylon inerme*			Antiproliferative activity with IC_50_ value of 46.8 ± 2.0 μg/mL	[165]
Ethanolic leaf extract of *Syzygium jambos*			Antiproliferative activity with IC_50_ value of 54.70 ± 0.60 μg/mL; inhibited cyclooxygenase-2 enzyme with IC_50_ value 3.79 ± 0.90 μg/mL	[168]
Ethanolic leaf extract of *Vanilla planifolia*			Antiproliferative activity with IC_50_ value of 31.2 μg/mL; induced DNA fragmentation	[169]
Methanolic aerial part extract of *Verbascum nigrum*			Antiproliferative activity with IC_50_ value of 81.92 μg/mL; however, fraction VNF4 (consisting of ilwensisaponins A and C, songarosaponins A and B) showed an IC_50_ of 12.27 μg/mL	[170]

**Table 9 molecules-26-01979-t009:** Plant derived bioactives and their biological effect on keratinocyte carcinoma.

Name/Bioative Ingredient	Source/Origin	Treatment	Outcome	References
**BCC**				
10% Sinecatechins ointment (Veregen^®^)/Epigallocatechin-gallate (EGCG)	*Camellia sinensis* L. leaf extract containing catechins (>85%)	Application to sBCC (39 patients)	No significant difference between placebo and treatment group Caused erythema, edema, erosions, crusts, and itching Insufficient uptake of active by sBCC cells in current formulation; different formulation should be considered	[171]
**SCC**				
0.5% Curcumin of 50% ethanolic turmeric extract in vaseline ointment	Polyphenolic present in rhizomes of *Curcuma longa* L.	Application to ulcerated tumor (62 patients)	Foul odour of wound reduced by >90% and itching was reduced Pain (potentially due to anti-inflammatory activity of curcumin), lesion thickness, and exudates from ulcer was reduced in 50%, 10%, and 70% of cases respectively One patient reported severe itching (possible due to curcumin allergy) Ethanol used to prepare the extract caused irritation, not present in an ointment In pre-clinical trials, xenografted mice, with SRB12-p9 SCC, showed significant suppression of tumor growth when treated with oral, topical, or combined	[172,173]
EGCG (6.5 µmol) once daily for 5 days a week (18 weeks in total)	Major catechin found in *Camellia sinensis* L.	Application to SCC tumors developed in female hairless SKH-1 mice irradiated with UVB for 20 weeks (twice weekly)	Reduced the number of non-malignant and SCC per mouse by 55% and 66%, respectively Tumor volumes reduced; however, SCC tumor size did not reduce	[174]
Caffeine (6.2 µmol) once daily for 5 days a week (18 weeks in total)	Methylxanthine alkaloid found in coffee		Reduced the number of non-malignant and SCC per mouse by 44% and 72%, respectively Tumor volumes reduced, however SCC tumor size did not reduce	[174]
Betulin-based Oleogel-S10	Pentacyclic triterpenes isolated from *Betula pubescens* Ehrh.	Application to patients with actinic keratosis (157 patients)	Once and twice a day application resulted in complete tumor clearance in 3.9% and 6.8% of patients, respectively Not significant clearance when compared to placebo	[175]

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
