# Peer review of "Pathogenesis of Keratinocyte Carcinomas and the Therapeutic Potential of Medicinal Plants and Phytochemicals"

_molecules, 2021, doi:10.3390/molecules26071979_

Round 1
Reviewer 1 Report
Major comments:
- It would be helpful to include a cartoon illustrating the epidermis layers, origins of BCC and SCC. It will be more visual and easier to understand.
- Similarly, it would benefit the reader if there is a world-wide map dictating the cases/incidence of BCC in different countries.
- Revise some titles. For example, the tile of session 10 is Squamous cell carcinoma and only contains 10.1.1. Squamous cell carcinoma in situ.
- Add a table to 15 to summarize the existing pharmacotherapies, including efficacy, advantage vs disadvantage for each treatment. Similarly, summary tables for sessions 16 and 17 are also helpful.
- Please simplify Tables 4 and 9. Too many sentences.
Author Response
- It would be helpful to include a cartoon illustrating the epidermis layers, origins of BCC and SCC. It will be more visual and easier to understand.
A figure has been inserted (Figure 1), which illustrates where squamous and basal cells are found within the epidermis layer of the skin.
- Similarly, it would benefit the reader if there is a world-wide map dictating the cases/incidence of BCC in different countries.
BCC cases are either under reported or completely left out of cancer registries globally. Section 4, paragraph 1, highlights that these are the reasons which contribute to the underestimation of skin cancer cases. Therefore, an image of a world map representing BCC cases will not correctly reflect the number of BCC cases, however a world map has been included (Figure 2) which shows the incidence of non-melanoma skin cancer as estimated by GLOBOCAN in 2020.
- Revise some titles. For example, the tile of session 10 is Squamous cell carcinoma and only contains 10.1.1. Squamous cell carcinoma in situ.
Revision and re-numbering of section titles have been done, as some sections have been converted into tables, as per the fourth query. Additionally, section 10 has been renumbered and now is “8. Squamous cell carcinoma”, with subheading “8.1. Squamous cell carcinoma in situ” and “8.2. Invasive squamous cell carcinoma (SCCI)”.
- Add a table to 15 to summarize the existing pharmacotherapies, including efficacy, advantage vs disadvantage for each treatment. Similarly, summary tables for sessions 16 and 17 are also helpful.
The section entitled “Topical pharmacotherapies currently used for the treatment of KC”, which was previously numbered section 15, is now section 11, and has been summarized into a Table (Table 6), representing efficacy, mechanism of action, disadvantages and side effects or existing pharmacotherapies.
The section entitled “Transdermal delivery of skin cancer drugs”, which was previously numbered section 16, is now a sub-section (12.3) under “Transdermal delivery of drugs” (Section 12) and has been summarized into a Table (Table 7).
The section entitled “Potential therapeutic effects of phytochemical/ medicinal plants against KC” which was previously numbered section 17, is now section 13, and has been summarized into a Table (Table 8), representing phytochemical/ origin, treatment and outcome. Additional phytochemical/ medicinal plants have been added to this section.
- Please simplify Tables 4 and 9. Too many sentences.
The Table entitled “Various subtypes of actinic keratosis “which was previously Table 4, is now Table 3, and has been simplified and reduced. Table 9 has also been simplified and reduced.
Reviewer 2 Report
Some subchapters are presenting limited amount of information on the topic and in certain instances it seems the essential information is missing (e.g., 14.1. Targeted delivery through the skin, or 14.2. Transdermal drug permeation routes). In some cases the information provided is not fully in line with current understanding or practice (e.g., Administration of drugs through the skin can follow two potential routes, trans-epidermal and trans-appendegeal, or These needles range from 25-200 microns in
length).
Author Response
- Some subchapters are presenting limited amount of information on the topic and in certain instances it seems the essential information is missing (e.g., 14.1. Targeted delivery through the skin, or 14.2. Transdermal drug permeation routes). In some cases the information provided is not fully in line with current understanding or practice (e.g., Administration of drugs through the skin can follow two potential routes, trans-epidermal and trans-appendegeal, or These needles range from 25-200 micronsin length).
The authors have added more information to the section entitled “Transdermal delivery of drugs” (Section 12), where “Targeted delivery through the skin” is sub-section (12.1), and “Transdermal drug permeation routes” is sub-section (12.2).
An article by Ramadon et al, 2021, also describes transdermal drug administration via two routes. Reference: Ramadon, D., McCrudden, M.T.C., Courtenay, A.J. et al. Enhancement strategies for transdermal drug delivery systems: current trends and applications. Drug Deliv. and Transl. Res. (2021). https://doi.org/10.1007/s13346-021-00909-6. Furthermore, an image (Figure 3) displaying these routes were added to sub-section 12.2.
Reviewer 3 Report
This review has a very broad focus on keratinocyte carcinomas, from incidence to diagnosis and treatment. My feeling is that it is overly broad. I would recommend to be focused on medicinal plants and phytochemicals for KC treatment and exclude from the final version the information about incidence and demographics, diagnosis, clinical variants etc. This logic is more appropriate for clinical guidelines, not for review for journal focused on pharmacology, drug discovery and medicinal chemistry. Incidence in South Africa can be a subject for separate review for relevant journal
Author Response
- This review has a very broad focus on keratinocyte carcinomas, from incidence to diagnosis and treatment. My feeling is that it is overly broad. I would recommend to be focused on medicinal plants and phytochemicals for KC treatment and exclude from the final version the information about incidence and demographics, diagnosis, clinical variants etc. This logic is more appropriate for clinical guidelines, not for review for journal focused on pharmacology, drug discovery and medicinal chemistry. Incidence in South Africa can be a subject for separate review for relevant journal.
The authors have minimized/ summarized information regarding incidence, variants, diagnosis, etc so as not to make the MS overly broad. These information has been kept in in the MS as Reviewer 1 has made some suggestions to keeping the information but summarizing it.
The existing topical pharmacotherapies (treatments) have been summarized into a Table (Table 6) and the mechanism of action summarized in order to minimize information about current treatments.
The section focusing on the incidence of non-melanoma skin cancer in South Africa has been removed, and a sentence has been added to the worldwide incidence only on the current South African statistics.
The section discussing the diagnosis of KC has been substantially reduced. This has also been merged with the tumour staging system in order to reduce content.
The section of clinical variants has been summarized (more information has been summarized in Table format) so as to reduce content.
Additional information on phytochemical/ medicinal plants effective against KC have been provided in Table 8.
Round 2
Reviewer 3 Report
I still feel that the topic is overly broad - my recommendation remain the same - to be focused on one topic only - use of medicinal plants and phytochemicals.
However, I do understand that in current format the review is also of great interest, and based on position of editor it can be published in it's current format.